# Relative Validity of a Short 15-Item Food Frequency Questionnaire Measuring Dietary Quality, by the Diet History Method

**DOI:** 10.3390/nu13113754

**Published:** 2021-10-24

**Authors:** Elisabet Rothenberg, Elisabeth Strandhagen, Jessica Samuelsson, Felicia Ahlner, Therese Rydberg Sterner, Ingmar Skoog, Christina E. Lundberg

**Affiliations:** 1Faculty of Health Sciences, Kristianstad University, SE 291 88 Kristianstad, Sweden; 2School of Public Health and Community Medicine, Institute of Medicine, Sahlgrenska Academy, University of Gothenburg, SE 413 90 Gothenburg, Sweden; elisabeth.strandhagen@gu.se; 3Centre for Ageing and Health (AgeCap), Department of Psychiatry and Neurochemistry, Institute of Neuroscience and Physiology, Sahlgrenska Academy, Sahlgrenska Academy, University of Gothenburg, SE 431 41 Gothenburg, Sweden; jessica.samuelsson@neuro.gu.se (J.S.); felicia.ahlner@neuro.gu.se (F.A.); therese.rydberg.sterner@neuro.gu.se (T.R.S.); ingmar.skoog@neuro.gu.se (I.S.); 4Cognition and Old Age Psychiatry Clinic, Psychiatry, Sahlgrenska University Hospital, Region Västra Götaland, SE 431 41 Gothenburg, Sweden; 5Department of Molecular and Clinical Medicine, Institution of Medicine, Sahlgrenska Academy, University of Gothenburg, SE 416 50 Gothenburg, Sweden; christina.lundberg@gu.se

**Keywords:** diet history, food frequency questionnaire, relative validity, dietary pattern, cardiovascular risk factors, screening tool

## Abstract

Food frequency questionnaires (FFQ) are commonly used dietary assessment tools. The aim was to assess the relative validity of a 15-item FFQ, designed for the screening of poor dietary patterns with a validated diet history (DH). The study population was derived from the Gothenburg H70 Birth Cohort Studies. The DH registrations were harmonized in accordance with the FFQ frequencies. The agreement was assessed by Cohen’s kappa with corresponding confidence intervals (CI) for the frequency and categorical variables. Bland–Altman plots were used for the numeric variables. The study comprised data from 848 individuals (55.2% women). Overall, there was high agreement between the methods, with the exact and adjacent level of agreement over 80% for eight variables. The proportion attributed to the opposite frequency was fairly low for most of the frequency variables. Most of the kappa values were in fair or moderate agreement. The highest kappa values were calculated for the type of cooking fat (k = 0.68, CI = 0.63–0.72) and sandwich spread (k = 0.55, CI = 0.49–0.53), and the lowest for type of bread (0.13, CI = 0.07–0.20) and sweets (0.22 CI = 0.18–0.27). In conclusion, the FFQ showed overall good agreement compared with the DH. We, therefore, think it, with some improvements, could serve as a simple screening tool for poor dietary patterns.

## 1. Introduction

Screening is considered as a quick and simple method aimed at identifying individuals at risk of an unhealthy condition [1]. Those identified at risk should be assessed by a more extensive method to confirm or exclude the actual condition.

Non-communicable diseases (NCDs), such as cardiovascular diseases, cancers, and diabetes, pose a great global health care challenge today. The risk of being affected by NCDs varies in the population, with low socioeconomic position as a strong predictor for poor health [2]. Most of the premature deaths are associated with such lifestyle factors as poor dietary patterns and low physical activity, risk factors that could be identified by screening. A large body of evidence shows that a healthy diet is a key factor for maintaining good health and preventing NCDs [3,4,5]. Such a diet is characterized by lower intakes of foods containing high levels of saturated fat and/or sugar combined with higher intakes of high-fiber foods [6].

However, according to the Nordic Monitoring System, the number of individuals with a healthy dietary pattern decreased by 20% between the years 2011 and 2014 [7]. In 2012, the Swedish National Board of Health and Welfare published the National Guidelines for Methods of Preventing Disease report [8], in which health-care personnel were urged to support patients in their efforts to improve their dietary pattern. Quick and easy screening methods are, therefore, needed to identify individuals at risk of poor dietary patterns. This is the rationale behind our previously developed short and simple food frequency questionnaire (FFQ) [9] aimed at indicating poor dietary patterns in relation to the recommended dietary pattern by the Nordic Nutrition Recommendations (NNR) [6].

The overall purpose of an FFQ is “A questionnaire in which the respondent is presented with a list of foods, and is required to say how often each is eaten in broad terms such as x times per day/per week/per month, etc. Foods included are usually chosen for the specific purposes and may not assess total diet” [10]. In the present case, the FFQ was aimed at screening the risk of poor dietary patterns [9]. The rationale was that health care personnel without specific knowledge within nutrition should be able to screen patients, for example, in a waiting room, within primary care. At the time this FFQ was developed, there were no similar tools available.

Our short 15-item FFQ has been tested in a feasibility study for its ability to predict dietary cardiovascular risk factors in a random sample from a healthy middle-aged population [9]. We concluded that the FFQ is able to predict cardiovascular risk factors. However, the FFQ has not been tested for validity. Dietary assessment tools are affected by measurement errors, and their accuracy needs to be assessed [11,12,13]. A validity control implies that a comparison is made with another method judged to be superior [14] or with other errors involved. However, there is no “golden” standard; only the relative validity of a new method can be assessed [15]. At the same time, dietary assessment tools with high accuracy are the key for identifying associations between diet and disease.

The aim of this study was to assess the relative validity of a short 15-item semi-quantitative FFQ, designed as a tool for screening the risk of a poor dietary pattern, with a validated diet history (DH) interview in a sample of 70-year-olds. In this study, a poor dietary pattern was defined as a dietary pattern not in accordance with the dietary pattern recommended by the NNR.

## 2. Materials and Methods

### 2.1. Participants

We used a population-based sample of 70-year-olds (born in 1944) from the Gothenburg H70 Birth Cohort Studies (H70), Sweden, conducted in 2014–2016. Participants were systematically selected based on specific birth dates, and 1203 participated (response rate 72.2%). The study design has previously been described in detail [16].

All participants without dementia were invited to participate in the dietary examination; 342 individuals did not participate due to various reasons (e.g., declined to participate, poor health, lack of time, or no response). There were N = 861 individuals who participated in the dietary examination [17]. Of the 861 who participated in the dietary examination, 848 completed the FFQ. In total, the study population comprised 848 individuals who completed both the FFQ and participated in the DH interview. In the H70 study, results from the DH show that dietary patterns have changed during the past five decades when comparing data from five different birth cohorts of 70-year-olds, with an increase in healthy foods, such as fruits, vegetables, and fiber-rich foods, in later-born birth cohorts [17].

Ethical approval was obtained from the Ethics Committee for Medical Research in Gothenburg, reference number 869-13. The tenets of the Declaration of Helsinki were followed, and informed consent was obtained from all participants.

### 2.2. Data Collection

#### 2.2.1. Food Frequency Questionnaire

The FFQ, found in the Appendix A, was originally designed as a quick screening tool to give an overall view of an individual’s dietary pattern and to identify poor dietary patterns with focus on risk factors for cardiovascular diseases [9]. It comprised questions regarding frequency of consumption of different food groups based on indicators from the NNR of a healthy dietary pattern [6]. Information on portion sizes is not provided. The FFQ was either completed during the day for the general examination, or the participants could complete it later at home and return it by mail.

#### 2.2.2. Diet History

A dietitian conducted a semi-structured face-to-face 60–90 min interview estimating food intake during the preceding three months. The interview took place either in the participant’s own home, or at an outpatient clinic. The interview included both structured and open-ended questions about usual food patterns in order to capture habitual food intake as equivalent as possible. The same DH method has been used in all H70 cohorts where dietary habits have been investigated [17]. It was performed as a semi-structured interview by a dietitian capturing habitual dietary intake (food and beverages) during the past 3 months. The interview was conducted at the outpatient clinic or during a home visit, included not only open-ended questions about usual food patterns but also structured questions in order to capture total intake as closely as possible. Data were processed using the Swedish National Food Agency’s nutrient database (The Swedish Food Composition Database) to estimate energy and nutrient intake. It has been described in detail previously [16,17]. The DH method has been validated and found to give comparable energy values to those predicted by the heart rate method, activity diary, and double labelled water, as well as by calculating the ratio between energy intake and basal metabolic rate (BMR) [18,19].

Dietary intake was registered as grams of food items usually consumed per day/week/month for calculation of individual intake. The participants’ reported intake of, in total, 1810 different food items divided into 35 food groups in accordance with a study from the Swedish Food Agency on dietary patterns [20] (the procedures have been described in detail previously [17]). Those of the 35 food groups applicable to the items in the FFQ were selected in further analyses and categorized in accordance with the options on the FFQ. The selected food groups and registered food intake in the DH are summarized in the Appendix A (Appendix A). Individual intake during the DH was summarized in frequencies and intervals (day/week/month), where each reported food item was considered as one intake of the applicable item on the FFQ. All intakes of each item were summarized as week total or daily total. As such, a reported consumption of, e.g., tuna once a week and salmon once a week in the DH were summarized as eating fish two times a week.

#### 2.2.3. Harmonization of DH Data vs. the FFQ

All items measuring frequencies in the FFQ were condensed from four to three levels of frequencies (see Appendix A) because the removed level was not considered to add any further information. This applies to question (Q)1 vegetables, Q2 fruit/berries, Q3 nuts, Q4 fish, Q5 red meat, Q6 white meat, Q7 sweets, and Q10 dairy products. Corresponding frequencies and intervals (day/week/month) reported during the DH were summarized as week total or daily total and coded as applicable to the answer options for the corresponding item in the FFQ.

In the FFQ, Q8 “How often do you eat breakfast” was dichotomized into every day (every day/almost every day) and not every day (a few times a week/once a week). In the DH “Having breakfast” was identified by an open-ended question on usual meal pattern. Respondents reported usual meal pattern in terms of type of meal and time for 24 h on a regular day and on weekends. As the interviewer did not explicitly ask for breakfast (since there could be a risk of directing the answer of the respondent), breakfast was defined as regular intake of a lighter meal between the hours 5.30 and 9.30 a.m. For example, if the participant only reported intake of one glass of orange juice during these hours, it was not defined as a breakfast.

Q9 (bread) was divided into two separate questions: one for number of slices and one for type of bread in data analysis. Q9a, frequency of slices/pieces of bread usually eaten per day in total, was summarized as daily total intake both in the FFQ and DH. Q9b, types of bread, were coded as white bread, whole wheat/crispbread, or combinations of above. All individuals who stated “other” in the FFQ (n = 60) were not included in the agreement analysis in accordance with category classification. All types of bread registered in the DH were summarized and coded in the same way. Those who got a summarized intake of less than once a week in the DH were classified as “does not eat bread” and were not included in the agreement analysis (n = 31).

Type of milk/sour milk/and yoghurt usually consumed (Q11) was coded as full fat (3%), semi-skimmed/reduced fat (1.5%), and skimmed/low-fat/non-fat (≤0.5%) milk both in the FFQ and DH. Type of sandwich spread (Q12) was coded as butter (>75% fat), margarine with plant sterols, and margarine (30–70% fat). Individuals who stated “does not use spread” or “does not know” were not included in the agreement analysis (n = 138 and n = 7 in the FFQ and DH, respectively). Frequencies of types of dairy products and sandwich spreads registered during the DH were summarized, and the most frequent category was selected. For individuals who had consumed two categories equally (n = 19), and, if one of these was in agreement with the option chosen in the FFQ, it was interpreted as agreement between methods.

Type of cooking fat (Q13) was coded as butter/margarine (60–80%), margarine with seed and plant oils/liquid margarine, and vegetable oil. Information on type of cooking fat in the DH was obtained with open-ended questions. Hence, the participants could state several options. For individuals who had stated two different types of cooking fat in the DH (n = 442), and when one of these was in agreement with the option chosen in the FFQ, it was interpreted as agreement between the methods. The option “does not use fat in cooking” and “does not know” was not included in the analysis (FFQ, n = 7; DH, n = 76).

Q14, on adding salt to food in the FFQ, was dichotomized into no (no/yes, sometimes) and yes (yes often/yes, I always add salt before I taste the food). Information on salt consumption during the DH was obtained using the open-ended question “do you usually add salt to your food” (during a meal). Q15 regarding avoidance of salt in the FFQ was not included in the present study as there were no equivalent questions in the diet history. Since the bread question was divided into two questions, the analysis ended up with 15 items.

#### 2.2.4. Participant Characteristics

Mean body mass index (BMI) with corresponding standard deviations (SD) was calculated based on measured height and weight values (kg/m^2^). Other characteristics of participants were dichotomized. Smoking was divided into current or non-smoker (never smoked or past smoker). Educational level was divided into compulsory primary school (≤9 years) or higher. Marital status was divided into married (currently married and/or cohabiting) or not married (never been married/not cohabiting, divorced, widowed). Country of birth was divided into Sweden or other.

### 2.3. Statistical Methods

Participant characteristics are presented as numbers and percent. The proportion of answers classified into the same, adjacent, or opposite category by both methods was calculated for the items: Q1–Q7 and Q10. In the results, this was expressed as percent of exact agreement (same frequency measured in both methods), adjacent agreement (frequency in FFQ was one step away from frequency measured in DH), or opposite agreement (frequency in FFQ was opposite of the frequency measured in DH). The proportion of answers classified into the same category for both methods was calculated for the items: Q8, Q9b, and Q11–Q14. In the results, this was expressed as percent of exact agreement between methods.

The agreement of individual classification between both methods was evaluated, and weighted or unweighted Cohen’s kappa values were calculated [21]. The Cohen’s kappa statistics indicate poor level of agreement for values under 0.20, fair level of agreement for values between 0.21–0.40, moderate level of agreement for values 0.41–0.60, good level of agreement for values between 0.61–0.80, and excellent agreement level of agreement for the values above 0.80 [22]. In addition, the sensitivity and specificity of the two dichotomous items were calculated (Q8, Q14).

Intake of number of slices of bread (Q9a) is presented as mean, standard deviation, percentiles, and as minimum and maximum. The agreement of this numeric variable was analyzed using a Bland–Altman plot, showing the mean differences of bread intake between the two methods along with 95% limits of agreement (LOA) [23]. This was conducted to graphically assess the presence of bias or disagreement. Data management and statistical analyses were performed in R version 3.6.2 (The R Project for Statistical Computing, Vienna, Austria).

## 3. Results

### 3.1. Participants

The present study included parallel data from the DH interview and FFQ in N = 848 participants (women n = 474, 55.2%). The background characteristics on the participants are presented in Table 1.

### 3.2. Level of Agreement

Table 2 shows a comparison of the distributions on each food item for the FFQ (columns) and DH (rows). Figure 1 shows the proportion of agreement between the methods for items measuring frequencies (Q1–Q7 and Q10). The exact and adjacent level of agreement was over 85% for all the items except for sweets, where the agreement was 78.7%. Hence, the proportion of answers attributed to the opposite frequency was fairly low for all the questions except for nuts, sweets, and dairy products. All the kappa values were of fair or moderate agreement, and the highest kappa values were calculated for fruit/berries (0.48, CI = 0.40–0.55), dairy products (0.48, CI = 0.41–0.54), and fish (0.44 CI = 0.38–0.50) (Figure 1).

Figure 2 shows the proportion of individuals with exact agreement between the methods for all the categorical items: Q8, Q9b, and Q11–Q14. The agreement was over 70% for breakfast, type of sandwich spread, type of cooking fat, and salt, but somewhat lower for type of bread and type of dairy products, with an exact level of agreement of 53.1% and 66.5%, respectively.

The kappa values for the categorical items varied from a poor to good level of agreement. The type of cooking fat k = 0.68 (CI = 0.63–0.72), sandwich spread k = 0.55 (CI = 0.49–0.60), and dairy products k = 0.48 (CI = 0.42–0.53) were all of moderate and good agreement, while the type of bread had the lowest level of agreement of k = 0.13 (CI = 0.07–0.20) (Figure 2). Both the breakfast and salt items had high sensitivity (0.95 and 0.83, respectively) but lower specificity (0.25 and 0.45, respectively) (Figure 2).

There was a statistically significant difference between the two means of slices of bread (Q9a). The Bland–Altman plot shows that the participants tend to underestimate intake through the FFQ compared with the DH. This means that there was a systematic underestimation bias on the reported slices of bread (Q9a) in the FFQ. The mean absolute difference in the bread intake between the methods was 2.93 (CI = 2.74–3.12) slices. The lower LOA was −2.65 (CI = −2.98–−2.32) and the upper LOA was 8.51 (CI = 8.18–8.84) (Figure 3).

## 4. Discussion

The present validation study has investigated the relative validity of a semi-quantitative 15-item FFQ aimed at screening the risk of poor dietary patterns. Data from the FFQ were compared with corresponding data from a DH, both used in a population-based sample of 70-year-olds. High agreement between the methods, with the exact and adjacent level of agreement over 80%, was found for a majority of the variables. The proportion attributed to the opposite frequency was fairly low for most of the frequency variables.

The agreement between the methods varied for different food groups. Good agreement between the methods, with high exact or adjacent agreement, was found for a majority of the frequency items, such as vegetables, fruit, fish, white meat, and dairy products. Good agreement was also found in most of the categorical items: breakfast, type of dairy products, type of sandwich spread, type of cooking fat, and salt use. Less good agreement was found for nuts, red meat, sweets, and type of bread. In addition, there was a systematic underestimation of the number of slices of bread, where participants on average underreported an intake of almost three slices.

Even though both methods rely on the participants’ memory, the context of how the dietary intake is reported by the FFQ and the DH differs significantly, and the respondents may think differently when they report the intake by the two methods. The period for reporting the actual intake also differs somewhat between the methods. The FFQ was aimed at a habitual dietary pattern without any time perspective, while the time perspective in the DH was a habitual dietary pattern within the last three months. There is a risk in dietary validations that the first method used could influence the participants’ answers in the second one since the reporting of dietary habits might make people aware of and reflect on their dietary intake, which, in turn, might affect the response to the second method. In the present case, the period between the participants’ responses on the two dietary methods was from about 2 weeks to some months apart. Therefore, we judge the problem as minor that the FFQ answers may have affected the responses in the DH. Furthermore, the FFQ is a short self-administered questionnaire, opposite to the DH, which is based on an extensive interview by a dietitian of the respondents in their own homes.

The DH-interviewer needs, of course, to be as neutral and objective as possible when asking questions not to influence the answers. However, it might be easier to remember and reflect on intake when a professional interviewer put the question in a structured way for about one hour and encouraged the respondent not to forget anything consumed. If the participant was unsure, e.g., regarding the type of bread consumed, the interviewer could ask to see the packaging. Especially for red meat, the discrepancy between the methods was considerable. This may be because the respondents in the FFQ had difficulty categorizing dishes correctly. In the interview, the dietitian was able to ask follow-up questions to find out what type of meat the respondent ate.

Regarding sweets, there was a considerable difference in the reported intake between the methods, which may be explained by the fact that, in the DH, the interviewer supported the participant to remember by control questions. Furthermore, the question regarding sweets in the FFQ was broad, including many different food items. According to dairy products, the difference between the methods might be explained by the fact that some people use milk products with different fat contents and they reported the most common alternative differently in the FFQ and DH, respectively. With regard to nuts, the difference between the methods could be explained by the fact that nuts were not eaten routinely in this cohort of older adults.

There is no consensus in the literature regarding which statistical method is the most suitable for assessing the validity of dietary tools. Analysis by Bland–Altman plots and Kappa analysis are both frequently reported [10,24,25]. Using more than one approach demonstrates the robustness of the validation process [10]. The classification capacities of the tools can be analyzed by comparing, through Kappa analyses and contingency tables, the concordance or agreement within the distribution by tertiles, quartiles, or quintiles. The results can then be presented as an exact agreement (classified in the same category by both methods), plus or minus one category, and gross misclassification [10]. The main advantage is that, with cross-classification, the percentages misclassified clearly illustrate the likely impact of measurement error. It has been established that 50% of the subjects correctly classified and <10% of subjects grossly misclassified into thirds, and weighted kappa values above 0.4 are desirable [25]. However, this seems to be difficult to achieve in the validation studies of the FFQ [24].

### 4.1. Future Adaption and Improvements of the FFQ

A systematic review of the validation of the semi-quantitative FFQ summarizes that the validity results are not always favorable for all the nutrients or food groups evaluated. Improvements of the tool are recommended [24]. The FFQ method has several limitations; taken together, this means that it is only able to give an indication of the quality of the dietary pattern in relation to the recommendations [6]. However, its simplicity is also its greatest strength as a quick and easy screening tool trying to find individuals at risk of poor dietary patterns, e.g., within primary care and preventive medicine. Within health care, screening for the risk of malnutrition is well established [26]. The common methods for malnutrition screening are the SGA (subjective global assessment) [27] and MNA (minimal nutrition assessment) [28,29]. However, screening for a poor dietary pattern as a risk factor for NCD is not well established, at least not in Sweden. Primary care and preventive medicine could benefit from a quick screening as a basis for a conversation with the patient about further steps for diet change, for example, dietetic counselling.

In this validation study, we have obtained valuable experience on how to improve the FFQ. First, the items on the FFQ, measuring frequencies were condensed from four to three levels. This change was made before any analysis because the levels did not add any further information as separate frequencies. This merger also enabled better and more robust analysis through kappa statistics. We believe that three levels of frequencies are sufficient to determine the dietary pattern and recommend that the levels are merged in further use. Second, there was a systematic underestimation of the reported slices of bread (Q9) that should be taken into consideration when interpreting an individual’s dietary pattern. We also observed some missing information on Q9b, number of breads, which might be explained by the duality of the question. Some respondents might have misunderstood the question because there is no option “does not eat bread” in the second step of the question, which is focusing on the type of bread. We suggest dividing this question into two separate questions, one for the number of slices and one for the type of bread, to prevent misunderstandings. Finally, regarding Q12 (spread) and Q13 (cooking fat), we suggest simplifying them by giving a lower number of alternatives. By these alterations, we think the conditions for this FFQ to serve as a simple screening tool will be improved.

### 4.2. Strengths and Limitations

The strength in the present study is the large homogeneous population-based sample in terms of the city of residence and age, which is an advantage when the aim is to compare the outcome from two different methods. Furthermore, dietary history is a well-established validated method, here considered as the reference method. However, there are some limitations related to the comparison between the methods. The data in the DH were not originally intended to be summarized in accordance with the frequencies in the FFQ. However, since the FFQ should be regarded as a screening instrument, not a detailed dietary examination, the calculated frequencies are rough estimations of the individual intake of food groups. Found “at risk” is not equal to having poor dietary habits. Therefore, individuals found at risk within primary care and preventive medicine might be referred to a dietitian for a more in-depth assessment and possibly consultation.

Information on the intake of polyunsaturated fat from cooking is missing, and the breakfast variable was derived by setting a time interval for the first meal during the day, which can be questioned as a method for classifying breakfast. A strength is that we have used different statistical methods, both Cohen’s kappa and Bland–Altman, aiming to present the robustness of the validation process [10]. However, the analysis through Cohen’s kappa has its limitations. Although the level of the exact and adjacent agreement between the methods was high for most items, some of the calculated kappa values indicated only a fair or moderate level of agreement. Cohen’s kappa is sensitive to uneven answer distributions and is, therefore, mostly used on quintile data. However, it was not possible to recalculate the frequencies into quintiles in the present study. In addition, the greater the expected chance agreement, the lower the resulting value of the kappa. Therefore, dichotomous variables are assigned lower estimated kappa values because the odds of ending up in one or the other category is 50%. This considered, we think that the agreement between the methods is higher than indicated by the fair or moderate level of agreement (see Figure 1).

## 5. Conclusions

We have validated a short 15-item FFQ against a DH in a population-based sample of 70-year-olds. The validation study has some limitations related to the FFQ and some related to the DH. There was good agreement between the methods, with the exact and adjacent level of agreement over 80% for a majority of the variables. The proportion attributed to the opposite frequency was fairly low for all the frequency variables, except nuts, sweets, and dairy products. Based on the present results, we think that the FFQ, with some improvements, could serve as a simple screening tool for poor dietary patterns.

## Figures and Tables

**Figure 1 nutrients-13-03754-f001:**
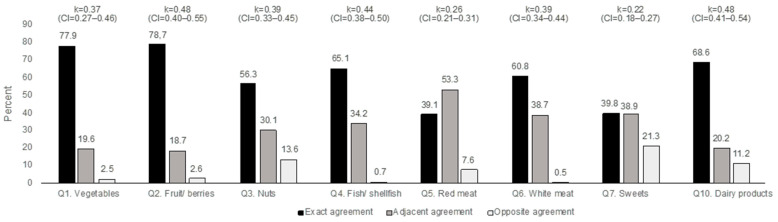
Proportion of agreement between methods for frequency items in the Food Frequency Questionnaire compared with Diet History, and respective weighted kappa values.

**Figure 2 nutrients-13-03754-f002:**
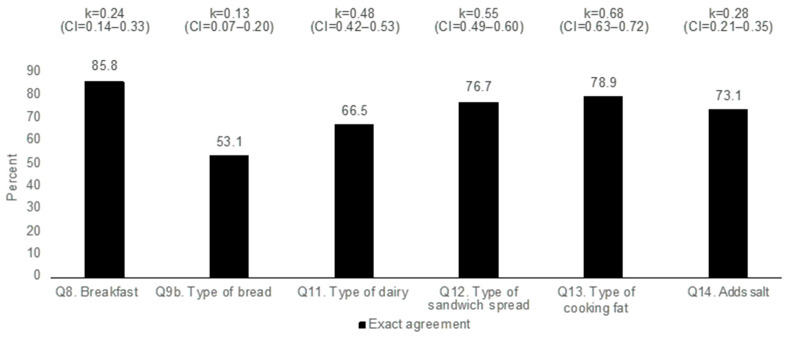
Proportion of agreement between methods for categorical items in the Food Frequency Questionnaire compared with Diet History, and respective weighted kappa values.

**Figure 3 nutrients-13-03754-f003:**
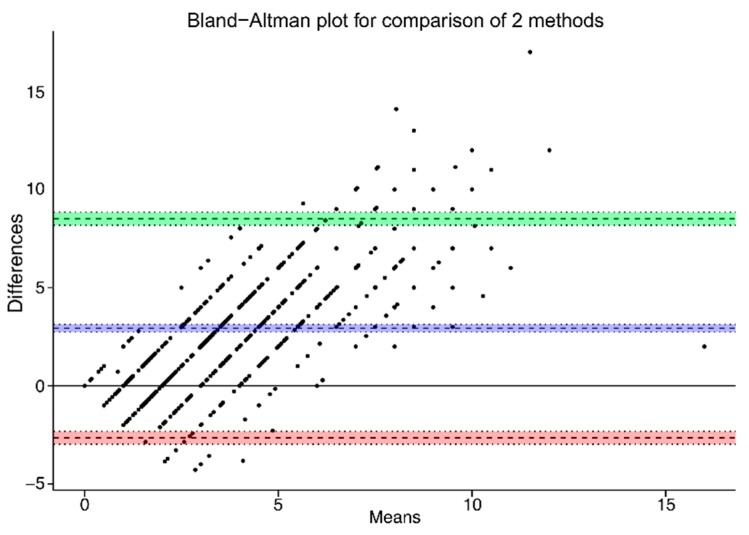
Bland–Altman plot of Q9a, agreement between number/slices of bread reported during the DH and the FFQ (n = 848). The purple/blue line represents the mean difference between the two methods, and the red and green lines represents the limits of agreements corresponding to ±1.96 SD (lower agreement: −2.65 slices/day, upper agreement: 8.51 slices/day).

**Table 1 nutrients-13-03754-t001:** Characteristics of participants.

n = 848	
Birth year	1944
Examination year	2014–2016
Sex, n (%) women	468 (55.2)
BMI ^1^, mean (SD)	25.9 (4.5)
Current smoker ^2^, n (%)	63 (7.4)
Educational level ^3^, n (%)	
<Primary school	150 (17.9)
Marital status ^4^, n (%)	
Married/Cohabitant partner	589 (69.7)
Country of birth (%)	
Sweden	757 (89.3)

BMI = body mass index; SD = standard deviation. ^1^ Missing on BMI, n = 11 (1.3%). ^2^ Missing on smoking, n = 2 (0.2%). ^3^ Missing on education, n = 8 (0.9%). ^4^ Missing on marital status, n = 3 (0.4%).

**Table 2 nutrients-13-03754-t002:** Comparison of distributions on each food item for FFQ (columns) and DH (rows).

Diet History (Rows)	Food Frequency Questionnaire (Columns)						
**Q1. How often do you eat vegetables?**							
	1 ^1^		2 ^1^		3 ^1^		Total
1. Once a day or more often	**612 (72.9) ^2^**		106 (12.6)		12 (1.4)		730 (87.0)
2. A few times a week	45 (5.4)		**38 (4.5)**		6 (0.7)		89 (10.6)
3. Once a week or less often	9 (1.1)		9 (18.1)		**10 (1.2)**		28 (3.3)
Total	663 (79.0)		152 (18.1)		24 (2.9)		847
**Q2. How often do you eat fruit and/or berries?**							
	1		2		3		Total
1. Once a day or more often	**609 (72.2)**		97 (11.5)		18 (2.1)		724 (85.8)
2. A few times a week	33 (3.9)		**44 (5.2)**		17 (2.0)		94 (11.1)
3. Once a week or less often	4 (0.5)		11 (1.3)		**14 (1.7)**		29 (3.4)
Total	645 (76.4)		151 (17.9)		48 (5.7)		847
**Q3. How often do you eat nuts?**							
	1		2		3		Total
1. Once a day or more often	**93 (11.0)**		56 (6.6)		35 (4.1)		184 (21.7)
2. A few times a week	22 (2.6)		**40 (4.7)**		18 (2.1)		80 (9.4)
3. Once a week or less often	80 (9.4)		159 (18.8)		**344 (40.6)**		583 (68.8)
Total	195 (23.0)		255 (30.1)		397 (44.9)		847
**Q4. How often do you eat fish or shellfish?**							
	1		2		3		Total
1. Three times a week or more often	**123 (14.5)**		207 (24.4)		6 (0.7)		336 (39.7)
2. Once or twice a week	30 (3.5)		**394 (46.5)**		30 (3.5)		454 (53.6)
3. A few times a month or less often	0 (0)		23 (2.7)		**34 (4.0)**		57 (6.7)
Total	153 (18.1)		624 (73.7)		70 (8.3)		847
**Q5. How often do you eat red meat?**							
	1		2		3		Total
1. Three times a week or more often	**173 (20.4)**		339 (40.0)		64 (7.6)		576 (68.0)
2. Once or twice a week	32 (3.8)		**121 (14.3)**		71 (8.4)		224 (26.4)
3. A few times a month or less often	0 (0)		10 (1.2)		**37 (4.4)**		47 (5.5)
Total	205 (24.2)		470 (55.5)		172 (20.3)		847
**Q6. How often do you eat white meat?**							
	1		2		3		Total
1. Three times a week or more often	**13 (1.5)**		48 (5.7)		3 (0.4)		64 (7.5)
2. Once or twice a week	31 (3.7)		**346 (40.8)**		46 (5.4)		423 (49.9)
3. A few times a month or less often	1 (0.1)		203 (23.9)		**157 (18.5)**		361 (42.6)
Total	45 (5.3)		597 (70.4)		206 (24.3)		848
**Q7. How often do you eat buns/cakes, chocolate/sweets, crisps or soda/juice?**							
	1		2		3		Total
1. Once a day or more often	**188 (22.2)**		189 (22.3)		172 (20.3)		549 (64.8)
2. A few times a week	22 (2.6)		**78 (9.2)**		108 (12.8)		208 (24.6)
3. Once a week or less often	8 (0.9)		11 (1.3)		**71 (8.4)**		90 (10.6)
Total	218 (25.7)		278 (32.8)		351 (41.4)		847
**Q8. How often do you eat breakfast?**							
	1			2			Total
1. Everyday	**701 (82.7)**			39 (4.6)			740 (87.2)
2. Not everyday	81 (9.6)			**27 (3.2)**			108 (12.7)
Total	782 (92.2)			66 (7.8)			848
**Q9b. What type(s) of bread do you eat?**							
	1		2		3		Total
1. White bread	**9 (1.2)**		6 (0.8)		9 (1.2)		24 (3.1)
2. Whole wheat bread/crispbread	3 (0.4)		**180 (23.3)**		228 (29.5)		411 (53.2)
3. Combinations of above	13 (1.7)		103 (13.3)		**221 (28.6)**		337 (43.7)
Total	25 (3.2)		289 (37.4)		458 (59.3)		772
**Q10. How often do you drink/eat milk, sour milk and/or yoghurt?**							
	1		2		3		Total
1. Once a day or more often	**453 (53.4)**		54 (6.4)		42 (5.0)		549 (64.7)
2. A few times a week	70 (8.3)		**42 (5.0)**		24 (2.8)		136 (16.0)
3. Once a week or less often	53 (6.3)		23 (2.7)		**87 (10.3)**		163 (19.2)
Total	576 (67.9)		119 (14.0)		153 (18.0)		848
**Q11. What type of milk, sour milk and/or yoghurt do you usually drink/eat?**							
	1		2		3		Total
1. Full fat (3%)	**158 (19.8)**		102 (12.8)		25 (3.1)		285 (35.8)
2. Semi-skimmed/reduced fat (1.5%)	52 (6.5)		**253 (31.8)**		45 (5.7)		350 (44.0)
3. Skimmed/low-fat/non-fat (<0.5%)	6 (0.8)		37 (4.6)		**118 (14.8)**		161 (20.2)
Total	216 (2.17)		392 (49.2)		188 (23.6)		796
**Q12. What kind of spread do you usually use on sandwiches?**							
	1		2		3		Total
1. Butter (>75% fat)	**383 (57.6)**		16 (2.4)		25 (3.8)		424 (63.8)
2. Margarine with plant sterols	9 (1.4)		**53 (8.0)**		47 (7.1)		109 (16.4)
3. Margarine (30–70% fat)	52 (7.8)		6 (0.9)		**74 (11.1)**		132 (19.8)
Total	444 (66.8)		75 (11.3)		146 (22.0)		665
**Q13. What kind of fat do you usually use for cooking at home?**							
	1		2		3		Total
1. Butter/margarine (60–80%)	**281 (34.1)**		64 (7.8)		18 (2.2)		363 (44.1)
2. Margarine made with seed and plant oils/liquid	37 (4.5)		**151 (18.3)**		37 (4.5)		225 (27.3)
3. Vegetable oil	13 (1.6)		5 (0.6)		**217 (26.4)**		235 (28.6)
Total	331 (40.2)		220 (26.7)		272 (33.0)		823
**Q14. Do you usually add salt to your food?**							
	1			2			Total
1. No/yes sometimes	**515 (61.8)**			109 (13.1)			624 (74.8)
2. Yes, often/always	115 (13.8)			**95 (11.4)**			210 (25.2)
Total	630 (75.5)			204 (24.5)			834

Data are presented as numbers and (percent). FFQ, food frequency questionnaire; DH, diet history; Q, question. ^1^ See row label for corresponding column label. ^2^ Bold letters are those categorized in the same box.

## Data Availability

The data presented in this study are available on request from the corresponding author. The data are not publicly available due to legal restrictions.

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
