# Peer review of "Relative Validity of a Short 15-Item Food Frequency Questionnaire Measuring Dietary Quality, by the Diet History Method"

_nutrients, 2021, doi:10.3390/nu13113754_

Round 1

Reviewer 1 Report

Rothenberg et al. Presented a very good paper that addresses the validation of a short 15-item food frequency questionnaire measuring dietary quality. I have a few observations:

  1. The authors stated the aim of the study was to assess the relative validity of a 15-item FFQ, designed for screening of poor dietary patterns with a validated diet history (DH).

However, the authors do not present results for the dietary patterns found in the studied population. In addition, they could compare the FFQ dietary pattern with the results from the validated diet history.

  1. The conclusion in the abstract does not match the conclusion of the paper “FFQ with some improvements could serve as a simple screening tool for poor dietary patterns.”
  2. One of the objectives of using the Bland Altman method is to detect if there is systematic over or underestimation bias. If a significant correlation between the average and the difference between both methods is found that means that there is a systematic bias. It would be interesting to know those correlations for Q9 to detect if there is a systematic bias between methods.

Author Response

Thank you for reviewing our manuscript. Please see attachment for our answers.

Reviewer 2 Report

This study investigated the relative validity of the newly developed FFQ with the diet history among 70-year-old seniors in Sweden. Although the study have potential merit for detecting poor dietary patterns as a risk of CVDs, there are several concerns need to be addressed.

#1 The introduction section needs to be comprehensively revised. The authors explained what the screening is, and the history for developing the FFQ. These statemants did not well justify the study aims; rather, what is FFQ (generally), why the FFQ have been developed and what were the advantages of the FFQ compared with other FFQs should be explained.

#2 What is the definition for “poor dietary pattern”? Does this mean the composite of excessive intake of nutrients (e.g., energy, carbohydrate, fat, or sodium) and/or shortage of nutrients (e.g., vitamins, dietary fiber, potassium, or phytochemicals)? Or does it simply mean not aligning with the NNR?

#3 This study only investigated the agreement between the FFQ and food frequency derived from the diet history. As this FFQ is designed for screening to “poor dietary patterns”, the author should justify the true “poor dietary patterns” is a risk factors for the CVDs or other adverse outcomes in targeted population.

#4 The study population in this study is quite different from the study that investigated the feasibility of the FFQ (50 years old vs. 70 years old). Could the FFQ truly generalise for the current study’s population? What was the rationale for utilizing the FFQ to older age group that might have different risk factor for mortality from younger people?

#5 Why the FFQ items were condensed from four to three levels? If the categorization of the each question was established arbitrarily, the results can be biased toward the study hypothesis. Please justify.

#6 As the FFQ provides overall score (0-15) and subsequent judgement for dietary patterns (3 categories), I suggest comparing the scores and categories deriving from the FFQ with DH, using kappa statistics. Discuss the results. 

Author Response

(The authors gave the same response as above.)

Round 2

Reviewer 2 Report

The manuscript have substantially improved. I have no further comment.

This manuscript is a resubmission of an earlier submission. The following is a list of the peer review reports and author responses from that submission.

Round 1

Reviewer 1 Report

Aim of this paper is to assess the relative validity of a short 15-item food frequency questionnaire, designed as a tool for screening risk of poor dietary pattern according to the Nordic Nutrition Recommendations.

This study has many similarities with work the same authors conducted a few years ago. https://pubmed.ncbi.nlm.nih.gov/30215342/

The authors should clarify:
1. what are the aspects of difference between this work and the previous one? 
2. what is the possible clinical usefulness of this short FFQ?

3. Is it a FFQ test that might be useful for particular populations e.g. assessing Malnutrition in institutionalised elderly? or might it only be useful in clinical trials?

4. what are the advantages of using this test compared to the traditional food history? what are the differences with the MNA test, which is currently the most widely used test in clinical practice?

mino points: pay attention to certain plagiarised parts (please see the attached file)

Author Response

Thank you for your valuable input!
Please see our answers below.

  1. What are the aspects of difference between this work and the previous one? 

    As we state in the introduction the present short FFQ is a new tool for screening for poor dietary habits. In the previous paper, we tested in a feasibility study for its´ ability to predict dietary cardiovascular risk factors. In the present paper, the aim is to assess the relative validity of the tool.

  2. What is the possible clinical usefulness of this short FFQ?

    The clinical usefulness is to serve as a simple screening tool for poor dietary patterns e.g. in patients with life style related risk factors. It is aimed for use in e.g. primary care and preventive medicine. This has been further clarified in the introduction and discussion.

  3. Is it a FFQ test that might be useful for particular populations e.g. assessing Malnutrition in institutionalised elderly? or might it only be useful in clinical trials?

    Please see reply 2.

  4. What are the advantages of using this test compared to the traditional food history? what are the differences with the MNA test, which is currently the most widely used test in clinical practice?

    The advantage of the present tool is that it is quick and easy to perform. Patients can fill in the FFQ in the waiting rooms themselves without the help of healthcare professionals. The DH takes just about an hour to complete and requires participation of specially trained staff, usually a dietitian.

Reviewer 2 Report

Dear Editor, the manuscript is interesting. It describes results of a study conducted to assess the relative validity of a short 15-item food frequency questionnaire with a validated diet history interview. Results are of interest for readers and add a significant contribution on the specific research field, since a validated shorth FFQ will be useful to assess nutritional habits of large populations. However, in order to improve the manuscript, I suggest the following amendments:

  • In the aim of the study authors should highlight that the FFQ has been validated in sample of 70-year-olds people and that the FFQ is a semi-quantitative tool since information on portion sizes is not recorded.
  • I suggest to start the discussion section with a brief summary of the main findings of the study and then a discussion of the results and a more critical judgement of the methodology. Finally, authors conclude that “Based on the present result we think that the FFQ with some improvements could serve as a simple screening tool for poor dietary patterns”, I suggest to give further details on this fundamental aspect.

Author Response

Thank you for your valuable input! Please see our answers below.
We have revised the manuscript in accordance with your suggestions (revisions shown with track changes).

1) (title) are you talking about a screener (method)? "Short FFQ" is ok, but "screener (method)" has been defined by FAO (http://www.fao.org/3/I9940EN/i9940en.pdf): "A retrospective dietary survey method that uses modified versions of longer Food Frequency Questionnaires varying in length, frequency categories and number of foods listed." It would be better to check and edit the title accordingly.

Many thanks for this information we did not know about this FAO document. In the FAO document, it is stated, “Consequently, screeners are used in situations when there is no need for comprehensive assessment. They are also used for surveillance, to screen individuals for inclusion in intervention or clinical trials, to identify and separate large numbers of individuals into groups or to distinguish individuals with low or high intakes.”.

The way we think of screening is defined in ESPEN Guidelines on Definitions and Terminology of Clinical Nutrition https://www.espen.org/guidelines-home/espen-guidelines. “Risk screening is a rapid process performed to identify subjects at nutritional risk”. The tool is aimed for clinical care e.g. primary care and preventive medicine were the term screening referrers to a quick method that should capture risks at an individual level a parallel to e.g. mammography screening but within the field of nutrition. Those identified at risk should go through a more in-depth assessment. We have now added a reference in the text to the definition of screening.

2) line 35: you also use the term "screening". Is it better to name it as screener rather than a short FFQ?

 Please see our reply on comment 1.

3) line 49-57: it is quite difficult to understand the flow of the introduction. This paragraph has rare information regarding your topic, while all the rest paragraphs were about the dietary survey method directly. Or you want to mention why questionnaire accuracy is important? if so, probably put this paragraph as the first paragraph of the introduction?

We agree and have now deleted that part of the introduction.

4) line 63 & line 196: it would be better to make them the same, for instance, rephrase the section 2.1 as "2.1 participants"

We have changed to “participants”.

5) line 79-82: you gave a definition of FFQ. The definition should be better put in the introduction part.

We have moved the definition as suggested to the introduction.

6) line 164: you kept describing statistical indicators, rather than the statistical methods. 

As suggested we have moved “participant characteristics” to a new sub heading 2.2.4. under "Data collection".

Reviewer 3 Report

This manuscript presented important research regarding the validity analysis of a dietary survey questionnaire. The analysis is logical and scientific, the results are well-presented, and all the tables & figured are very clear. However, the terminology and flow of the manuscript still require authors' attention for further editing.

1) (title) are you talking about a screener (method)? "Short FFQ" is ok, but "screener (method)" has been defined by FAO (http://www.fao.org/3/I9940EN/i9940en.pdf): "A retrospective dietary survey method that uses modified versions of longer Food Frequency Questionnaires varying in length, frequency categories and number of foods listed." It would be better to check and edit the title accordingly.

2) line 35: you also use the term "screening". Is it better to name it as screener rather than a short FFQ?

3) line 49-57: it is quite difficult to understand the flow of the introduction. This paragraph has rare information regarding your topic, while all the rest paragraphs were about the dietary survey method directly. Or you want to mention why questionnaire accuracy is important? if so, probably put this paragraph as the first paragraph of the introduction?

4) line 63 & line 196: it would be better to make them the same, for instance, rephrase the section 2.1 as "2.1 participants"

5) line 79-82: you gave a definition of FFQ. The definition should be better put in the introduction part.

6) line 164: you kept describing statistical indicators, rather than the statistical methods. 

Author Response

Thank you for your valuable input! Please see our answers below.
We have revised the manuscript in accordance with your suggestions (revisions shown with track changes).

  • In the aim of the study authors should highlight that the FFQ has been validated in sample of 70-year-olds people and that the FFQ is a semi-quantitative tool since information on portion sizes is not recorded.

The aim has been amended according to suggestion.

  • I suggest to start the discussion section with a brief summary of the main findings of the study and then a discussion of the results and a more critical judgement of the methodology. Finally, authors conclude that “Based on the present result we think that the FFQ with some improvements could serve as a simple screening tool for poor dietary patterns”, I suggest to give further details on this fundamental aspect.

We have developed the text in the start of the discussion aimed as a brief summary.

We made some amendments in the text (subheading 4.1) and added some sentences to clarify how the tool should be understood and how it could be used in healthcare (subheading 4.2).

We also have given details regarding how the FFQ could be improved, but now tried to rephrase this part to be clearer.

Round 2

Reviewer 1 Report

I thank the authors for attempting to improve the paper. Unfortunately, I did not notice much difference with the first version of the study. The reported critical issues remain:
- beyond the 80% concordance between DH and 15-FFQ, replacing the accuracy and professionalism of the dietitian's assessment with a simple test of a few questions needs much more convincing evidence. 

-the discussion of the paper is superficial, there are no nutritional evaluations but only statistical comments. 

-- the validated diet history is not described, it is difficult to understand how the comparison with FFQ was made

- too many self-citations make the work lose the credibility it needs to be published.

Reviewer 3 Report

no more comments.